# Formation of Nano- and Micro-Scale Surface Features Induced by Long-Range Femtosecond Filament Laser Ablation

**DOI:** 10.3390/nano12142493

**Published:** 2022-07-20

**Authors:** Joerg Schille, Jose R. Chirinos, Xianglei Mao, Lutz Schneider, Matthias Horn, Udo Loeschner, Vassilia Zorba

**Affiliations:** 1Laserinstitut Hochschule Mittweida, University of Applied Sciences Mittweida, Technikumplatz 17, 09648 Mittweida, Germany; schneide@hs-mittweida.de (L.S.); horn2@hs-mittweida.de (M.H.); loeschne@hs-mittweida.de (U.L.); 2Lawrence Berkeley National Laboratory, Berkeley, CA 94720, USA; jrchirinos@lbl.gov (J.R.C.); x_mao@lbl.gov (X.M.); vzorba@lbl.gov (V.Z.); 3Department of Mechanical Engineering, University of California at Berkeley, Berkeley, CA 94720, USA

**Keywords:** filament, femtosecond laser, laser ablation, LIPSS, ripple, surface texturing, long-range propagation, filament wandering, rainbow color

## Abstract

In this work, we study the characteristics of femtosecond-filament-laser–matter interactions and laser-induced periodic surface structures (LIPSS) at a beam-propagation distance up to 55 m. The quantification of the periodicity of filament-induced self-organized surface structures was accomplished by SEM and AFM measurements combined with the use of discrete two-dimensional fast Fourier transform (2D-FFT) analysis, at different filament propagation distances. The results show that the size of the nano-scale surface features increased with ongoing laser filament processing and, further, periodic ripples started to form in the ablation-spot center after irradiation with five spatially overlapping pulses. The effective number of irradiating filament pulses per spot area affected the developing surface texture, with the period of the low spatial frequency LIPSS reducing notably at a high pulse number. The high regularity of the filament-induced ripples was verified by the demonstration of the angle-of-incidence-dependent diffraction of sunlight. This work underlines the potential of long-range femtosecond filamentation for energy delivery at remote distances, with suppressed diffraction and long depth focus, which can be used in biomimetic laser surface engineering and remote-sensing applications.

## 1. Introduction

The formation of **l**aser-**i**nduced **p**eriodic **s**urface **s**tructures, also referred to as LIPSS or ripples in the literature, is a well-known phenomenon occurring when polarized laser beams irradiate solid-state surfaces [1,2,3,4,5,6]. LIPSS are characterized by crests and valleys that run perpendicular to the incident electrical field. Two types of ripple are distinguished in the literature, according to in their characteristic spatial period: (I) *LSFL* ripples, denoting **l**ow-**s**patial-**f**requency **L**IPSS with a period near the wavelength of the irradiating laser beam, and (II) *HSFL* ripples, denoting **h**igh-**s**patial-**f**requency **L**IPSS as a kind of periodic nano structure with a period considerably smaller than the laser wavelength. In a very recent publication asking ten open questions about LIPSS [7], it was made apparent that the formation mechanism of ripple structures is still a topic of scientific discussion. Different theories were presented in the past, including self-organization effects, optical interference, and the scattering of the incident radiation at surface defects, surface plasmon polariton excitation, and hydrodynamic processes [1,2,3,4,5,6,7,8,9,10,11,12,13,14,15,16,17,18]. All these complex physical processes affect the coupling efficiency of incident laser radiation, which may result in periodic energy distribution and modulated material removal to finally form periodic surface reliefs. Starting from nano-scale surface features originating after the first pulse irradiation, the transition to periodic ripple structures is driven by positive-feedback mechanisms [8,11,14,15,16,17,18] when preceding pulses interact with the altered superficial layer, which continually changes from pulse to pulse in the multi-pulse regime. The effects of multi-pulse feedback phenomena, particularly the formation of the *LSFL* ripples of larger spatial period, are the focus of the present study. For a linearly polarized laser beam irradiating a metal surface, the spatial period *Λ* of the *LSFL* ripples is given by:(1)Λ=λRe [η] ± sinθ                   with g ‖ E
where *λ* is the incident laser wavelength, *θ* is the angle of the incidence, *η* = [*ε*_d_*ε*_metal_/(*ε*_d_ + *ε*_metal_)]^1/2^ is the effective refractive index of the dielectric-metal interface for surface plasmons, *ε*_d_ is the dielectric constant of the ambient medium, *ε*_metal_ is the dielectric constant of the metal, Re [*η*] is the real part of *η*, g is the grating vector, and *E* is the tangential component of the electrical field vector of the incident laser beam [13,19]. In addition to the wavelength, the unique LIPSS features can be controlled by the fluence, polarization state, pulse number, etc., of the impinging laser beam.

Femtosecond lasers have become commonly used tools in micro/nano-manufacturing, and have emerged as attractive sources for ripple texturing for numerous applications in the field of modern surface engineering and functionalization [13,19,20,21,22,23,24,25]. This is because of their unique advantages in materials processing, such as the suppression of thermal diffusion, nonlinear multiphoton excitation of carriers, deterministic optical breakdown threshold, internal modification of transparent materials, and reproducible nanoscale resolution [26,27]. Recently, the use of femtosecond filaments has emerged as an alternative means to process materials due the intense clamping inside the filament core, which allows beams to propagate with suppressed diffraction. This characteristic enables energy delivery at standoff or remote distances and provides a very long depth of focus, which can be leveraged in laser materials processing as well as remote-sensing applications based on laser ablation and plasma generation (e.g., laser-induced breakdown spectroscopy, LIBS, and laser ablation molecular isotopic spectrometry, LAMIS).

Femtosecond filamentation is the result of a complex, dynamic balance between optical Kerr self-focusing and plasma-defocusing effects experienced by intense laser beams undergoing nonlinear propagation. The laser intensity inside the core of the air filament is clamped, along the beam path, to an almost constant value in the range ~10^13^–10^14^ W/cm^2^ [28,29,30]. The filament core has a diameter of ~100 μm, while the surrounding beam, commonly termed the energy reservoir, can be much larger. As the filament propagates in a self-guiding manner, the beam-energy reservoir continuously replenishes the energy expended inside the filament core to various nonlinear loss mechanisms [29]. Control of the filament initiation distance can be achieved by the manipulation of both the pulse duration and the assistance of beam-launching optics [31,32,33].

The irradiation of a sample with femtosecond filaments is greatly relaxed and independent of its position within the filament path. As a consequence, filament-based laser ablation provides a major advantage for the processing of curvilinear or humped surfaces compared to common sharp-focusing femtosecond laser beams, as materials ablation and structure development can take place without being affected by the depth of focus. Initial studies reported the manufacturing of nano- and microscale structures that allowed the strong absorption of broadband light waves on samples with spherical Al surfaces [34]. Furthermore, short filaments (3–4 cm long) enable processing for the fabrication of a broadband low-reflectivity black silicon surface by ablating crystalline silicon [35,36].

The goal of the present study is to investigate the formation of laser-induced periodic surface structures formed on stainless steel with femtosecond-filament launched at long distances. Specifically, we explore the femtosecond-filament-induced LIPSS properties between 50 m and 55 m from the laser source, as well as their properties across different locations within the filament path. SEM and AFM analysis methods were employed to evaluate the characteristic of the laser filament-induced microscopic surface features. The quantification of the periodicity of the filament-induced self-organized surface structures was accomplished by discrete two-dimensional fast Fourier transform analysis (2D-FFT). Furthermore, we studied the effects of the number of filament laser pulses and the pulse-to-pulse pointing stability on the distribution of self-organized structures formed on the sample target upon interaction with the filament. This work provides new insights into the filament-induced LIPSS at propagation distances that have not been studied to date. These findings also demonstrate the basis for remote surface texturing, manufacturing, and optimizing laser–matter interactions and energy coupling in remote-sensing applications.

## 2. Filament Laser Processing and Topography Analysis Methods

### 2.1. Laser Filamentation

Filamentation of femtosecond laser pulses propagating in a medium can be accomplished if their peak power is higher than a critical power:(2)$$P_{cr}=\frac{3.77\lambda^{2}}{8\pi \;n_{2}n_{0}},$$
where *n*_0_ and *n*_2_ are the linear and Kerr refractive index (3.01 × 10^−19^ cm^2^/W), respectively, and *λ* is the wavelength of the laser beam. There is no universal agreement about the exact value of the critical power for filamentation in air, which is estimated here following Equation (2) at *P_cr_* = 3.2 GW. Efficient control of the filament generation and propagation is essential for several technologies, including laser-induced breakdown spectrometry (LIBS) [37,38] and laser-ablation molecular isotopic spectroscopy (LAMIS) [39,40] for remote sensing at long distances. The nominal distance for filament generation *Z*_fil_ is given by the Marburger formula:(3)$$Z_{\mathrm{fil}}=\frac{0.367\;kr^{2}}{\sqrt{{\left(\sqrt{\frac{P}{P_{cr}}}-0.0852\right)}^{2}-0.0219}},$$
where *r* is the beam radius, *P* is the peak power of the pulse, and *P_cr_* the critical peak power for filamentation. Since the self-focusing distance of a femtosecond pulse scales quadratically with the beam radius, it is common practice to magnify the beams with a beam expander to delay the collapse distance. Furthermore, the beam-expander approach provides a simpler and more robust means to fine-tune the ablation distance, compared with solely tuning the pulse duration through the introduction of negative chirp. When a beam expander involves two lenses, the effective filament initiation distance becomes:(4)$$Z_{\mathrm{fil}}^{F_{1},F_{2}}=d+F_{2}\frac{\;Z_{\mathrm{fil}}\left(F_{1}-d\right)-dF_{1}}{Z_{\mathrm{fil}}\left(F_{1}+F_{2}\right)+F_{1}F_{2}-d\left(Z_{\mathrm{fil}}+F_{1}\right)},$$
where *F*_1_ and *F*_2_ are the focal lengths of the lenses, and *d* is the distance between the two lenses. The focusing power of the system must be dynamically controllable by moving the optics with a translation stage to achieve efficient fine-tuning of the filament-initiation distance.

### 2.2. Experimental Setup

The study of long-distance laser-filament processing was performed by using a femtosecond laser system (Astrella, Coherent Inc., Santa Clara, CA, USA) delivering a Gaussian beam with 800-nanometer wavelength, 5.2 mJ of maximum pulse energy, and 120-femtosecond pulse duration (measured by FROG analysis). The maximum peak power of the laser beam was *P*_0_ = 41.0 GW, which is about one order of magnitude higher than the critical power for filament formation. Following equation 5, the peak power is calculated from the pulse energy *Q*_P_ divided by the pulse duration *τ*_H_ and corrected by a temporal-pulse-shape-dependent constant factor, which is 0.94 for Gaussian pulses
(5)$$P_{0}=0.94\times \frac{Q_{P}}{\tau_{H}}.$$

The position of the laser filament within the beam path was controlled with the help of a beam expander, as shown Figure 1. The onset of filamentation represents the position in the beam path where we first start to observe the first sign of ablation on a polyvinyl chloride (PVC) sample. For the experiments described in this work, the filament initiation distance was adjusted to 50 m away from the laser source. At this distance, however, we did not observe any ablation on the stainless-steel sample, which can be attributed to the different optical and thermophysical properties between PVC and stainless steel.

The filament initiation distance of 50 m from the laser-source-beam exit is hereafter referenced as “0 m—position” in the filament path. Accordingly, a 51-meter beam-propagation distance is referred to as “1 m—position”, which means within 1 m of the propagating filament; 52-meter distance is referred to as “2 m—position” within the filament; 53-meter distance is referred to as “3 m—position” etc. Limited by the free laboratory space, the longest distance between the substrate and the laser source was 55 m, referred to as the “5 m—position” within the filament, even though the filamentation length was much longer. This is because the experimental system includes reflecting mirrors to fold the beam prior to the onset of filamentation, at which point special care has to be taken to avoid potentially exceeding the critical mirror-damage threshold. Due to the experimental setup design and laboratory’s space limitations, in this work, we focused on the first 5 m of filamentation, starting at a beam propagation distance of 50 m though 55 m from the laser source.

The mirror-polished stainless-steel metal sheets (AISI 304, 1 mm thick) were irradiated under stationary irradiation conditions. The substrates were placed at different positions within the propagating laser beam between 51 m and 55 m from the laser source, which corresponds to the 1 m-to-5 m position in the filament. The number of pulses irradiating the substrate surfaces varied between 1, 2, 5, 10, 20, 50, and 100. The pulse-repetition rate was kept constant at 1 Hz.

### 2.3. Analysis Methods

The effect of pulse number and substrate position on the micro-structural characteristics, spatial uniformity, and global homogeneity of the filament-induced surface features was analyzed within the filament path. First, SEM micrographs were taken from the filament-textured spots at 5000× and 7500× magnification using a JSM-6510 LV (JEOL Ltd., Tokyo, Japan) scanning electron microscope. Secondly, in order to evaluate the filament-produced surface textures, cutout sections from the micrographs were subjected to a discrete two-dimensional fast Fourier transform analysis (2D-FFT) following an approach described elsewhere [6,8,15,41,42]. The cutout sections were taken from the 7500× micrographs with dimensions of 12 × 12 µm^2^ and 0.01-micrometer spatial pixel resolution. The 2D-FFT images provided the spatial frequency of the quasi-periodic features from a surface in the Fourier space, where the spatial *LSFL* ripple period *Λ*_LSFL_ is inversely proportional to the wave number *k*, following:*Λ*_LSFL_ = 1/*k*.(6)

The SEM images displayed diverse periodically shape-repeating structures affected by the specific irradiation conditions, such as pulse number and substrate position, in the filament path. However, the evaluation of the laser filament-textured surfaces in this study was focused on the *LSFL* ripples. Accordingly, the spatial ripple periods were calculated following Equation (6) from the respective spatial frequencies *k*, calculated from the corresponding Fourier transform images.

In addition, the **d**ispersion in the **L**IPSS **o**rientation **a**ngle (DLOA) δ*θ* [6,42] was determined from the SEM micrographs. The DLOA describes the distribution of the local LIPSS orientation. It is used as a representative measure of homogeneity and angular orientation of the femtosecond-filament-induced surfaces features. The DLOA was determined by implementing the OrientationJ plugin, which is available through the open-source image-processing software ImageJ. In accordance with recently published work [42], the Riesz Filter was applied as structure sensor, Gaussian smearing was not included, and the DLOA distribution was corrected from its offset.

Finally, the depth of the microscopic surface features was evaluated by AFM measurements to provide more information about the topographical characteristics of the laser-filament-produced surface textures.

## 3. Results and Discussion

### 3.1. Filament Pointing Stability

Figure 2 shows photomicrographs of the laser-filament-ablated spots produced with increasing numbers of pulses on the AISI 304 substrates. The substrate position was fixed 55 m from the laser source, which corresponds to the 5-meter position in the filament path. In Figure 2a, there is no overlap between the ablation zones from two pulses, indicating the transverse fluctuation of the filament from pulse to pulse on the substrate. This filament-pointing fluctuation, also referred to as filament wandering in the literature, can be attributed to atmospheric or filament-heating-induced turbulences, as well as natural perturbations of the refractive index in the laboratory-air environment [43,44,45,46]. In addition to these parameters, when discussing the pointing stability in our experiments, we have to also take into account the vibrations of the multiple mirrors at long distances (e.g., at 18 m and 36 m) used to bounce the beam back and forth in order to obtain long-distance propagation over the finite laboratory space. These vibrations can be attributed to inherit drift, air disturbance, transmission-medium variation, mechanical vibration, and elastic deformation [46,47]. These sources can disturb the wavefront of the propagating filament, with the consequence that both the onset/formation and transverse displacement of the filament are affected, which, in turn, causes filament fluctuation as a stochastic process. Accordingly, with the assumption of isotropic turbulences in the beam path, the transverse fluctuation of the filament, in the horizontal and vertical direction to the plane perpendicular to the propagation direction, followed a normal distribution, described by the Rayleigh distribution law [44]. This can be observed in Figure 2b,c, where it is shown that many impinging pulses interacted in the center of the ablated area. Furthermore, the diameter of the ablation zone increased with higher pulse number. This resulted from the stochastic characteristic of filament fluctuation, leading to random isotropic displacements from the beam propagation axis.

The full diameter of the ablation area produced with 100 pulses was measured at 1.58 mm, Figure 2c, which is much wider than the diameter of approximately 460 µm of the ablation area produced with the single-filament pulses, such as those outlined in Figure 2a,b by gray dotted circles. From these measurements, we concluded that the maximum transverse displacement of the filament centers at the 5-meter filament propagation length is around 560 µm, which is in the range of values found in the literature [43,44]. The pointing stability calculated at 55 m by taking into account the maximum displacement is less than 10 µrad, which is indicative of a stable filament when compared with the reported values for continuous-wave laser beams stabilized by a beam-pointing control system [47].

Figure 3 presents the diameter of the ablation spots as a function of the substrate’s position in the propagating filament for different pulse numbers. The diameter of the ablation zone increased with increasing pulse number. Furthermore, the measurements show a wider diameter for the ablation zones produced at larger propagation distances. For example, the diameters were 1.13 ± 0.12 mm and 1.32 ± 0.11 mm for 50 and 100 pulses at the 2-meter position in the filament (52 m from laser source), enlarging to 1.37 ± 0.08 mm and 1.58 ± 0.05 mm at the 5-meter position (55 m from laser source), respectively.

In addition, the position of the substrate in the filament also affected the diameter of the individual ablation spots produced with single-pulse irradiations. Accordingly, a detailed view of the inset of Figure 3 shows the 354 ± 15-micrometer ablation spot diameter for the 2-meter position in the filament, increasing to 454 ± 26 µm at 5 m. These differences in the ablation spots’ sizes resulted from the comparatively weak ablation observed at 2 m, which steadily increased to stronger ablation at the 4-meter and 5-meter filament positions (corresponding to 54 m and 55 m of beam-propagation length).

### 3.2. Effect of Pulse Number on Surface-Texture Characteristics

Figure 4 shows SEM micrographs and the 2D-FFT analysis depicting the development of the surface texture in the center of the laser-filament-induced ablation spot. The SEM micrographs presented in the upper part of the figure show the evolution of the superficial surface texture in the center of the ablation spots as a function of the number of pulses. The ablation spots were produced by placing the substrate surface at the 5-m filament position (55 m from the laser source), where the strongest filament–matter interaction was observed in this study. Irregular nano-sized features developed randomly on the metal surface after processing with a single filament pulse. The nano-features are similar in size and shape to the nanostructures produced by tightly focused ultrashort-pulse laser beams, which are categorized in the literature into, for example, nano-pores, nano-craters, nano-bumps, nano-protrusions, nano-rims, etc. The size of the nano-features increased with the ongoing laser filament processing and, further, periodic ripple structures started to form in the ablation-spot center after the irradiation of five spatially overlapping pulses. With further increases in the pulse number, the regularly rippled surface texture changed into a rough and corrugated topology, with the surface predominantly covered with irregular molten spikes, bumps, and micro-holes. From this experiment, we concluded that the micro-structural characteristic of the developing surface texture was greatly affected by the number of spatially overlapping filament pulses.

The effect of the number of pulses on the homogeneity of the originating surface structures was examined in greater detail by 2D-FFT analysis (lower part of Figure 4), based on the SEM micrographs taken from the center part of the laser-filament-textured areas. The plot for the single-pulse ablation (left) indicates the irregular nature of the nano-features, while a high-regular structure with a period *Λ* = 662 nm (which is close to the wavelength of the laser beam) can be observed on the plot of the 20 pulses. The plot on the right-hand side, however, validates the irregularity of the surface textures produced with 100 pulses by a widely spread fragmented 2D-FFT map.

The AFM analysis revealed a clear surface roughening of the laser-filament-processed areas. This is evidenced, for example, in Figure 5 for the 5-m substrate position in the filament path (55-m beam propagation). Specifically, the depth of the nano-scale features produced by a single pulse was up to 50 nm (Figure 5, center), while the unprocessed polished metal surface showed an undulating cross-section profile (Figure 5, left). With higher pulse numbers, the feature depth increased between 50 nm to 100 nm for 20 pulses, as shown in Figure 5 (right). In addition, the ripple period can be estimated, from the 20-pulse cross-section profile, at about *Λ* = 653 ± 5 nm, which further validates the results of the 2D-FFT analysis presented above.

In addition to the applied pulse number, the position of the substrate in the filament path influenced the original surface texture. This is emphasized in Figure 6, showing SEM micrographs of the surface textures produced at different substrate positions along the filament path and the pulse numbers in the center of the respective ablation spots. The first appearance of nano-featured surface textures was observed at the 2-m position of the substrate in the filament path for 10 pulses, while ripples started to form at this distance after 20 pulse irradiations. At a shorter distance between the substrate and the onset of the filament, corresponding to the zero-position at 50-m beam-propagation length, the filament ablation was too weak to modify or texture the metal plate surface. At the longer filament-propagation distances, on the one hand, the ripples emerged at a lower pulse number. Hence, regular ripple formations originated after the irradiation of 10 and 20 pulses at the 4-m and 5-m substrate positions in the filament path (54 m and 55 m of beam-propagation length). At shorter distances, nano-scale surface features started to grow at these pulse numbers. On the other hand, with higher pulse numbers irradiating the substrate at 4 m and 5 m in the filament path, the sample surface was covered with irregularly disordered and molten features. This is shown in Figure 6, for 100 irradiating filament pulses. By applying 100 pulses at a shorter distance, however, a homogeneous rippled surface texture was obtained, as shown for the 2-m and 3-m filaments. Nevertheless, due to filament fluctuation and beam wandering, the number of pulses hitting the same spot area is a statistical value and the effective overlapping pulse number may vary. As a consequence, the characteristic feature topography of the laser-filament-produced textures varied depending on the locally prevailing irradiation conditions. This was observed, first, across the surface of the filament-ablation spots and, second, from one filament-ablation spot to the other. This is demonstrated in the following subsection.

### 3.3. Surface Textures Developing across the Ablation-Spot Surface

The top panel of Figure 7 presents the nano-craters, nano-groves, and nano-spikes formed at the outer areas of the ablation spot. This resulted primarily from the low number of spatially overlapping pulses in these areas, which was caused by the wandering of the filament on the substrate surface at these long-range distances. *LSFL* ripples with a mean period of *Λ* = 660 nm started to grow in the ablation-spot center with five pulses at the 5-m position in the filament path (55 m of beam propagation). At this centered position, as discussed above, there was a greater probability of a higher number of pulses hitting the same areas.

The effective number of overlapping filament pulses per spot area also affected the developing surface texture when a higher pulse number was applied. This is evident as the specific features varied across the ablation spot, something that could be observed independently from the irradiated pulse number. For example, as shown in the middle panel of Figure 7 for 20 pulses at 5 m (55 m beam propagation), regular ripples formed at the outer areas of the ablated spots. At these positions, the effective number of pulses was sufficiently high for *LSFL* ripple formation. By contrast, a merely disordered and rougher surface texture developed in the center of the ablation spots, resulting from the statistically higher effective pulse number in the center of the filament–sample interaction area.

The typical morphologies of the microscopic surface textures can also be observed on the ablation spots produced at different substrate positions in the filament. In a similar manner, the different types of microscopic surface features emerged in diverse areas within the ablation spot, influenced by the respective irradiating effective pulse number. However, small variations in the feature characteristics and the number of pulses needed for the formation of a desired microstructure were observed for surfaces that were irradiated with identical pulse numbers at different substrate positions in the propagating filament. This emphasizes the previous finding that the ablation performance depends on the position of the irradiated surface in the filament path. Here, the ablation was weaker at shorter filament-propagation lengths. The influence of the substrate position on the surface features became clear, for example, when comparing the SEM micrographs presented in Figure 7, middle vs. Figure 7, bottom. The displayed surface textures were produced with 20 pulses irradiated 5 m and 4 m from the onset of the filament in the beam path, respectively. In fact, the *LSFL* appeared less pronounced at the 4-m substrate position. Furthermore, these surfaces were covered by molten substructures, which, in turn, reduced the quality of the surface textures in terms of periodicity, homogeneity/regularity, and sharpness.

### 3.4. 2D-FFT Analysis for Quality Assessment of Laser-Filament-Produced Surface Textures

Figure 8 and Figure 9 show SEM micrographs of periodic ripple textures observed at 4-m and 5-m substrate position in the filament path. For each micrograph, detailed information on both the irradiation conditions and the exact position of the selected section within the wider ablation area are provided, such as *54 m, 10 P, 3 o’clock*, using the analogy of a 12-h clock to describe the 4-m substrate position in the filament (54-m beam propagation) for 10 filament pulses, and 3 o’clock as the relative position of the SEM-captured area within the spot. The SEM micrographs were evaluated by a 2D-FFT analysis to assess the spatial frequency of the laser-filament-made ripples. In the Fourier space, represented by a 2D-FFT map given below each SEM micrograph, the spatial frequency of the *LSFL* ripple texture is indicated by dominant sickle-shaped features. In addition, the DLOA is presented in the 2D-FFT maps to provide a measure of the regularity of the ripple texture. Notably, irradiation with a higher number of filament pulses at the 4-m substrate position is associated with a transition of the DLOA to narrower angles. For example, as presented in Figure 8, the DLOA decreased from δ*θ* = 30.5° for 10 pulses to δ*θ* = 12.9° for 100 pulses. The decreasing DLOA with higher numbers of overlapping pulses indicates the evolution of the filament-induced surface texture from a more curved pattern to a highly periodic grating of straight lines. Accordingly, this finding confirms that feedback mechanisms were also involved in the ripple formation under femtosecond-filament pulse irradiation, which is in line with the widely accepted theory of ripple formation described above. A minimum of 10 pulses per spot is required to produce uniform *LSFL* ripples with the filament. The ripple textures most likely originated in the outer areas of the filament-irradiated spots. Furthermore, in Figure 8 (bottom row), the wavenumber traces along the *k*-axis provide the spatial frequency of the ripple texture perpendicular (red) and parallel (green) to the laser beam polarization direction. For the filament texture produced at 4-m substrate position, the average value of the resulting wavenumber peak was *k* = 1.67 ± 0.08. This corresponded to *Λ* = 599 ± 29 nm ripple period, although the period of the *LSFL* ripples decreased notably with the increasing number of pulses.

Similarly, the surface textures produced at the 5-m substrate position in the filament (55 m beam propagation) are shown in Figure 9. For comparison, an inhomogeneous and fragmented surface texture developing in the center of the ablation spot after the irradiation of 100 filament pulses is shown in Figure 9 (right). Here, too, an important finding is that the most-regular *LSFL* ripples emerged in the outer regions of the filament-ablated spots. However, the *LSFL* ripples developed with high regularity at the 5-m substrate position, even at low numbers of overlapping pulses, which is clearly different from the 4-m position in the filament. This was validated by the small DLOA, which varied only slightly in the range between 12.0° < δ*θ* < 13.1° for 20 and 100 pulses. A potential explanation could be the stronger ablation at the 5-m substrate position in the filament, which is further supported by the tendency toward higher ripple regularity for pulses of higher fluence [42]. The average wave number and corresponding ripple period were determined as *k* = 1.61 ± 0.09 and *Λ* = 625 ± 34 nm, respectively. This is additional proof of the stronger filament ablation at the 5-m compared to the 4-m position, which is in agreement with the fact that the ripple period increases for higher fluence, as recently reported for the ultrashort pulse-laser ablation regime [48].

The ripple period and DLOA for a variety of highly regular and homogeneous laser-filament-produced surface textures are presented in Figure 10. The overall value of the ripple period was *Λ* = 627 ± 35 nm, which is within the range of *Λ* ~ 0.8∙λ. The DLOA was between 11.3° < δ*θ* < 30.5°, with smaller DLOA for the *LSFL* ripples produced at the 5-m than at the 4-m substrate position. This indicates that ripples with higher regularity could be produced at the 5-m position in the filament. The averaged DLOA for the *LSFL* ripples with the highest regularity was δ*θ* = 12.4° ± 0.8°. This value is also confirmed through the data referenced in the literature, which are within the range of 9.2° < δ*θ* < 15.0° [42]. The high regularity of the laser-filament-produced *LSFL* ripples was further verified by the rainbow-color effect, in which the uniform ripple texture acted as a diffraction grating (Figure 10, right). The coloring effect became visible for the first time on the three top row ablation spots, each produced with 10 filament pulses. This confirms the requirement of at least 10 pulses to form highly regular ripple textures, as exemplified by diffraction of sunlight.

## 4. Conclusions

A systematic study on the formation of ripples, or LIPSS, originating from the long-range femtosecond-filament-laser irradiation of stainless steel is presented for the first time. Both the number of filament pulses and the substrate position varied at substrate distances between 50 m and 55 m from the laser beam source, while the onset of the filament was adjusted at 50 m. The micro-structural characteristic of the filament-made surface textures was greatly affected by the effective number of spatially overlapping pulses. Initially, the nano-scale surface features developed irregularly after a few filament pulse irradiations, while periodic ripples started to form in the ablation-spot center after the irradiation of five filament pulses. The 2D-FFT analysis of the SEM micrographs captured from the filament-irradiated areas’ homogeneous LIPSS formed at the outer ablation spot areas with a ripple period in the range of *Λ* = 627 nm, which is close to the wavelength of the femtosecond laser beam. In addition, the smallest dispersion of the LIPSS orientation angle was determined as δ*θ* = 11.3° in the 2D-FFT analysis, which indicates highly regular LIPSS. The clear roughening of the laser-filament-processed surfaces was also detected in the AFM measurements, which showed 50–100-nanometer feature depths for the areas textured with 20 filament pulses. The high regularity of the filament-induced LIPSS was verified through the observation of the rainbow-color effect yielded by the angle-dependent sunlight diffraction. This work provides new insights into filament–matter interactions at long beam-propagation distances, and opens up new possibilities for the use of filaments in laser-processing applications in which a long depth of focus is needed (e.g., rough or curved surfaces), as well as in remote detection technologies requiring high-energy beam delivery for ablation and plasma generation.

## Figures and Tables

**Figure 1 nanomaterials-12-02493-f001:**
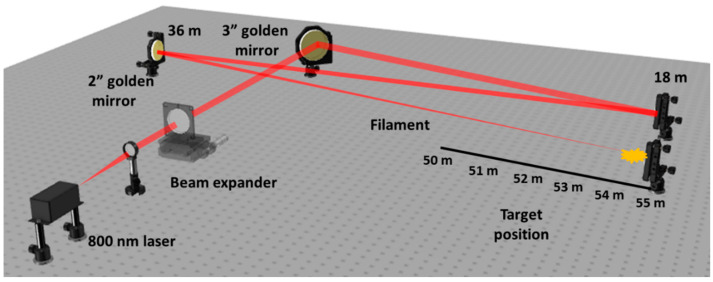
Experimental setup to launch the filament at long distances. Filament-induced LIPSS were studied across the range of 50–55 m from the laser exit. The laser beam was folded a few times before the filament was formed using gold-coated mirrors to reach the 55-meter target position inside the laboratory.

**Figure 2 nanomaterials-12-02493-f002:**
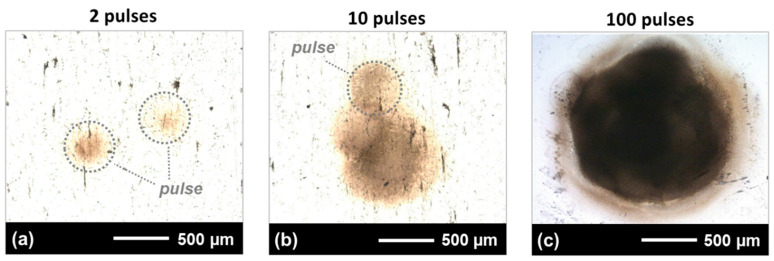
Optical microscopy images showing the femtosecond-filament beam-pointing stability on the substrate surface when the substrate was 55 m from the laser source (approximately 5-meter filament distance), as a function of number of pulses. The number of irradiated laser-filament pulses in this case varied between 1 and 100 (**a**–**c**).

**Figure 3 nanomaterials-12-02493-f003:**
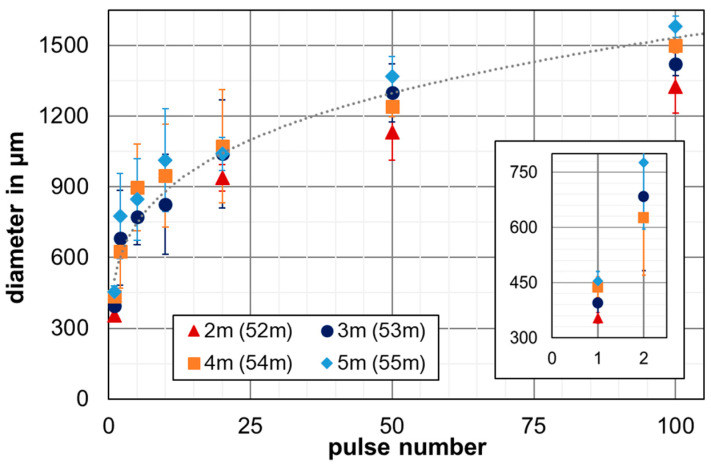
Increase in the filament-ablation-spot diameter as a function of pulse number and substrate position in the filament, which resulted from beam fluctuation in the propagating filament. The insert shows the ablation-spot diameters for 1- and 2-filament pulses in a magnified view.

**Figure 4 nanomaterials-12-02493-f004:**
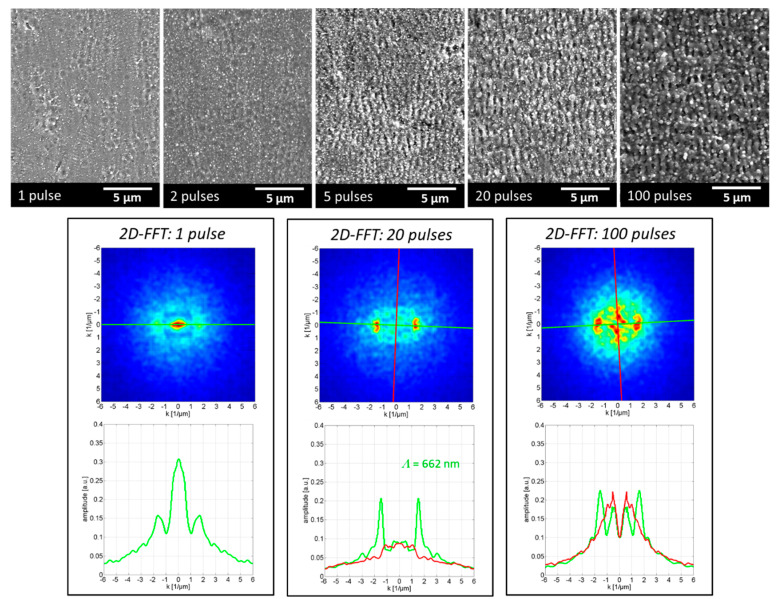
SEM micrographs and 2D-FFT analysis showing the development of the surface texture in the center of the laser-filament-made ablation spot at 5-meter substrate position in the filament path (55-meter beam propagation). The number of filament pulses varied.

**Figure 5 nanomaterials-12-02493-f005:**
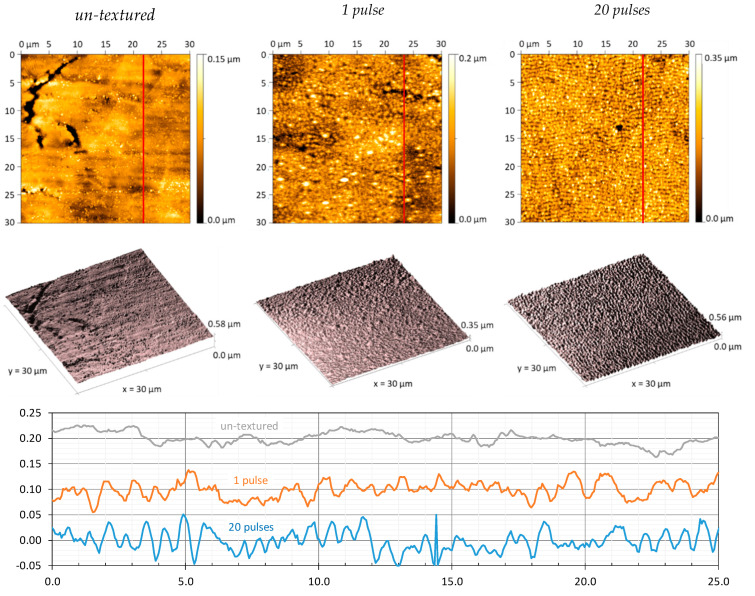
AFM analysis of un-textured (**left**) vs. laser-filament-textured steel surfaces achieved by irradiating 1 (**center**) and 20 (**right**) filament pulses at 5-m substrate position in the filament path (55-m beam propagation). The positions of the cross-section profiles shown in this figure’s lower part are indicated by red lines in the 2D height maps presented in upper part of the figure.

**Figure 6 nanomaterials-12-02493-f006:**
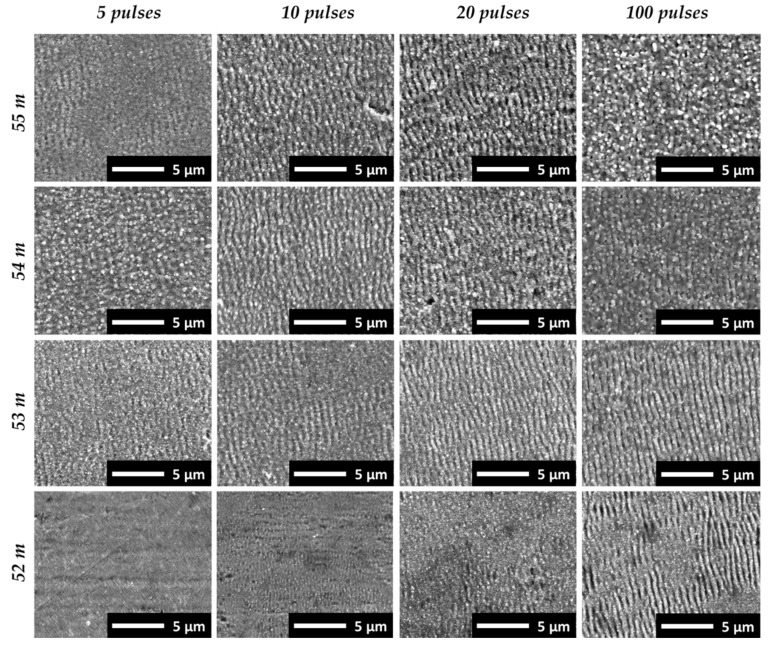
Overview of the surface textures developing in the center of the laser-filament-induced ablation spots at 2–5-m substrate positions in the filament propagation path and 52–55-m corresponding beam-propagation distances. The pulse number and substrate position in the filament varied.

**Figure 7 nanomaterials-12-02493-f007:**
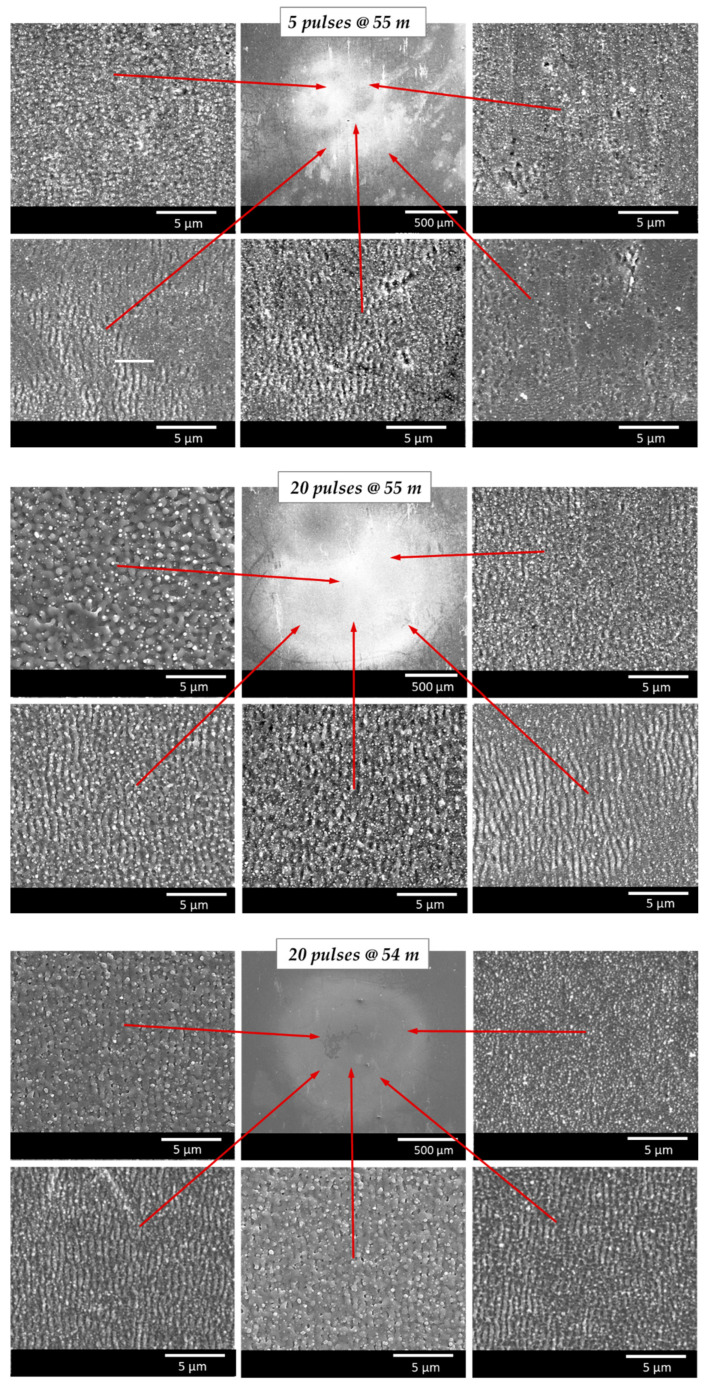
SEM micrographs showing the surface textures developing at different positions within the laser-filament-induced ablation spots (5 m (**top**, **center**) and 4 m (**bottom**) in filament path, or 55 and 54 m of total beam-propagation distance). The number of irradiated filament pulses and the position of the substrate surface in the filament varied.

**Figure 8 nanomaterials-12-02493-f008:**
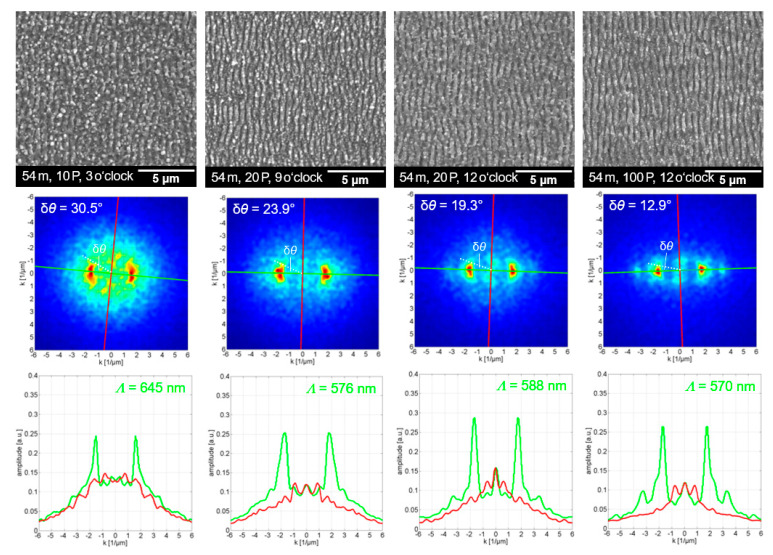
SEM micrographs (**upper** row), 2D-FFT map (**middle** row), and wave number profile along the *k*-axis (**bottom** row), as obtained in the outer areas of laser-filament-induced ablation spots at 4-m substrate position (54-m beam propagation).

**Figure 9 nanomaterials-12-02493-f009:**
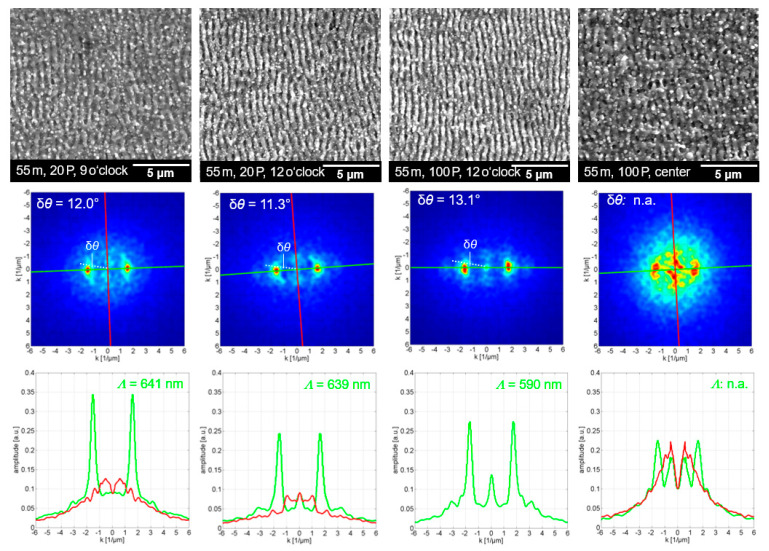
SEM micrographs (**upper** row), 2D-FFT maps (**middle** row), and wave-number profiles along the *k*-axis (**bottom** row), as obtained in the outer areas of laser-filament-made ablation spots at 5-m substrate position (55-m beam propagation).

**Figure 10 nanomaterials-12-02493-f010:**
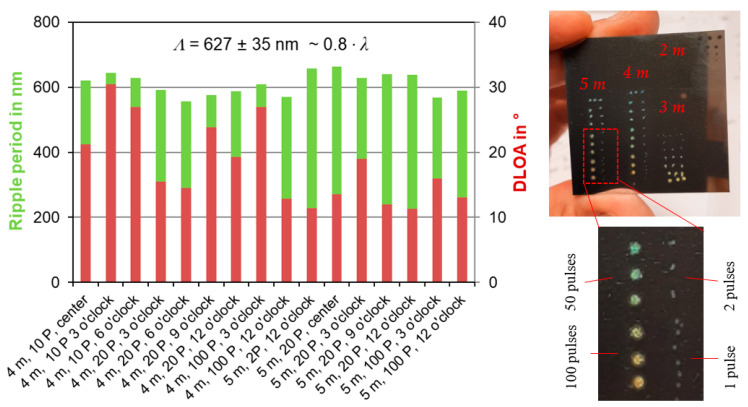
(**Left**): Ripple period and DLOA determined in the outer areas of laser-filament ablation spots produced at 4-m and 5-m substrate positions in the filament (54 m and 55 m of beam propagation); the number of irradiated filament pulses varied. (**Right**): Filament-made ripple texture as a diffraction grating for producing rainbow colors from sunlight.

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
