# Peer review of "Formation of Nano- and Micro-Scale Surface Features Induced by Long-Range Femtosecond Filament Laser Ablation"

_nanomaterials, 2022, doi:10.3390/nano12142493_

Round 1
Reviewer 1 Report
Authors of manuscript presented very intresting work with a goal of the investigation the formation of laser-induced periodic surface structures formed on stainless steel with femtosecond filament. They provide new insights into the filament-induced LIPSS at new propagation distances.
Minor remarks:
Abbreviations: LSFL, LIPPS should be introduced
In lines 34-43 there is a lot of information, but should as many as 16 works be cited there?
Line 58, reference 18 ([13, 18] should be changed to 17
In "dispersion in the LIPSS orientation angle (DLOA)" why some letters are underlined?
Figure 10 should be reformatted as they overlap
Author Response
Dear reviewer,
thank you for reviewing the manuscript and your valuable comments. Please find our answers in the attachement.

Reviewer 2 Report
Recommendation: major Revision
Comments:
In my view this may be considered for publication after the following modifications:
1. The comparison of the other similar research with this work in a table.
2. It is better take SEM with bigger magnification to show better morphology
3. What is the application for irradiated sample?
4. What is the irradiation cost of sample in dimension 2×3 cm?
5. Please add some new ref.(2021 and 2022)
6. The Conclusions section should include:
* A highlight of your hypothesis, new concepts and innovations.
* A summary of key improvements compared to findings in literature [give references to indicate
key improvements].
* Your vision for future work
Author Response

(The authors gave the same response as above.)

Reviewer 3 Report
The manuscript entitled “Formation of nano and micro scale surface features induced by long-range femtosecond filament laser ablation” of Joerg Schille et al have produced and characterized laser-induced periodic surface structures (LIPPS) at stainless steel surfaces using a long focal length mirror. The number of laser pulses and sample position were changed to see the effect on LIPPS formation and features. The main purpose of this work is to propose a configuration where the long-range focus provide a versatile design to induce LIPPS at big and irregular objects. In order to achieve the threshold for LIPPS formation, laser filamentation configuration was proposed. In my opinion, this manuscript can be accepted as it is. This work has good potential of interest for a broad audience.
Author Response
Dear reviewer,
thank you for reviewing the manuscript and your positive feedback on our research work.

Round 2
Reviewer 2 Report
Accept in present form